# Xpert MTB/RIF Ultra versus Xpert MTB/RIF for diagnosis of tuberculous pleural effusion: A systematic review and comparative meta-analysis

**Ashutosh Nath Aggarwal** *, **Ritesh Agarwal, Sahajal Dhooria** , **Kuruswamy Thurai Prasad, Inderpaul Singh Sehgal, Valliappan Muthu**

Department of Pulmonary Medicine, Postgraduate Institute of Medical Education and Research, Chandigarh, India

* aggarwal.ashutosh@outlook.com

## Abstract

### Objective

We compared diagnostic accuracy of pleural fluid Xpert MTB/RIF (Xpert) and Xpert MTB/RIF Ultra (Ultra) assays for diagnosing tuberculous pleural effusion (TPE), through systematic review and comparative meta-analysis.

### Methods

We searched PubMed and Embase databases for publications reporting diagnostic accuracy of Xpert or Ultra for TPE. We used bivariate random-effects modeling to summarize diagnostic accuracy information from individual studies using either mycobacterial culture or composite criteria as reference standard. We performed meta-regression through hierarchical summary receiver operating characteristic (HSROC) modeling to evaluate comparative performance of the two tests from studies reporting diagnostic accuracy of both in the same study population.

### Results

We retrieved 1097 publications, and included 74 for review. Summary estimates for sensitivity and specificity for Xpert were 0.52 (95% CI 0.43–0.60, $I^2$ 82.1%) and 0.99 (95% CI 0.97–0.99, $I^2$ 85.1%), respectively, using culture-based reference standard; and 0.21 (95% CI 0.17–0.26, $I^2$ 81.5%) and 1.00 (95% CI 0.99–1.00, $I^2$ 37.6%), respectively, using composite reference standard. Summary estimates for sensitivity and specificity for Ultra were 0.68 (95% CI 0.55–0.79, $I^2$ 80.0%) and 0.97 (95% CI 0.97–0.99, $I^2$ 92.1%), respectively, using culture-based reference standard; and 0.47 (95% CI 0.40–0.55, $I^2$ 64.1%) and 0.98 (95% CI 0.95–0.99, $I^2$ 54.8%), respectively, using composite reference standard. HSROC meta-regression yielded relative diagnostic odds ratio of 1.28 (95% CI 0.65–2.50) and 1.80 (95% CI 0.41–7.84) respectively in favor of Ultra, using culture and composite criteria as reference standard.

**Data Availability Statement:** All relevant data are within the paper and its Supporting Information files.

**Funding:** he authors received no specific funding for this work.

**Competing interests:** The authors have declared that no competing interests exist.

## Conclusion

Ultra provides superior diagnostic accuracy over Xpert for diagnosing TPE, mainly because of its higher sensitivity.

## Introduction

Tuberculosis (TB) is an important cause for exudative pleural effusions, especially in high TB burden settings [1]. However, a definite diagnosis of tuberculous pleural effusion (TPE) may often prove difficult. As TPE is a paucibacillary disease, mycobacterial culture positivity from pleural fluid samples is uncommon [1]. Pleural biopsy shows the typical caseating granulomatous inflammation, or even mycobacteria, more frequently in these patients. However, biopsy is an invasive procedure and hence still not routinely performed, especially in resource-constrained situations. Adenosine deaminase and interferon gamma are two commonly used surrogate pleural fluid biomarkers to diagnose TPE. Although both demonstrate good diagnostic accuracy for identifying TPE, there are wide variations in the assay techniques, and a uniform threshold is still not defined for either test [2, 3].

Xpert MTB/RIF (hereafter referred to as Xpert) was developed as a novel automated cartridge-based nucleic acid amplification assay to improve TB diagnosis with a short turnaround test time. Using a hemi-nested real-time polymerase chain reaction to amplify mycobacterial *rpoB* gene, the assay demonstrated improved sensitivity for identifying both pulmonary and extra-pulmonary TB [4, 5]. The most recent version, Xpert MTB/RIF Ultra (hereafter referred to as Ultra), attempts to further improve the limit of mycobacterial DNA detection by amplifying two different insertion sequences (IS*6110* and IS*1081*) in a larger reaction chamber [6]. Both insertion sequences are present in multiple copies only in *Mycobacterium tuberculosis* complex but not in other mycobacteria. Compared to Xpert, Ultra has shown a higher sensitivity, and marginally lower specificity, for diagnosing both pulmonary and extra-pulmonary TB [4, 5].

The World Health Organization (WHO) currently conditionally recommends Xpert as an initial diagnostic test for TPE, with moderate certainty of evidence [7]. Although pleural fluid Xpert assay is a promising tool for diagnosing TPE, its sensitivity is lower than that for some other forms of extra-pulmonary TB [8–10]. It is not clear whether the diagnostic accuracy of Ultra is significantly superior to that of Xpert for TPE. A recent systematic review could not identify enough studies for directly comparing the diagnostic performance of the two tests in pleural fluid [5]. We conducted this systematic review and performed independent meta-analyses to indirectly compare the diagnostic accuracy of both Xpert and Ultra, using both mycobacterial culture and composite clinical criteria as reference standards. We also directly compared the accuracy of the two tests from studies evaluating both tests in the same patients.

## Methods

We pre-registered the protocol for this review with PROSPERO registry (registration number CRD42021259421). Prior approval from our Institutional Ethics Committee was not necessary as we acquired summary information from already published articles. We report our findings according to the Preferred Reporting Items for Systematic Reviews and Meta-Analyses (PRISMA) guidelines [11, 12].

## Search strategy

We queried the PubMed and EMBASE databases for publications indexed till May 31, 2021. We used the following free text search terms: (Tuberculosis, Tuberculous, Tubercular, TB, Mycobacterial, Mycobacterium); (GeneXpert, Xpert, MTB/RIF, Ultra, Cepheid); and (Pleura, Pleural, Pleurisy, Pleuritis, Extra-pulmonary, Extrapulmonary, Non-respiratory, Nonrespiratory) for this purpose. If needed, we contacted investigators of selected publications for additional information. We also examined bibliographies of the included studies, as well as recent review articles, for any additional publications relevant to our analysis.

## Study selection and data extraction

After eliminating duplicates, two reviewers (ANA and RA) independently assessed all titles and abstracts identified from our literature search. We excluded animal research, studies on non-tuberculous diseases, publications not primarily reporting on diagnosis of TPE, conference abstracts, case reports, letters to editor not describing original observations, review articles, and editorials. The full texts of publications considered potentially eligible by either reviewer were further retrieved for more detailed evaluation.

We included a study for analysis if it (a) included patients with TPE and at least another cause of exudative pleural effusion, (b) used a microbiologic (mycobacterial culture positivity from pleural fluid or pleural biopsy), pathologic (granulomatous inflammation or presence of acid-fast bacilli on pleural biopsy), and/or clinical (overall clinico-radiological features and pleural fluid investigations suggestive of TPE, or favorable response to empiric anti-tubercular treatment) reference standard for diagnosing TPE, and (c) provided numerical data on sensitivity and specificity of Xpert or Ultra in TPE diagnosis using an appropriate reference standard. If the same patients contributed to diagnostic accuracy estimates in more than one study, only the publication examining the largest dataset was selected. In case of any disagreement, consensus between the two reviewers determined study inclusion.

We extracted the following information from studies finally included: study location, study design, patient inclusion and exclusion criteria, clinical and demographic characteristics of patients studied, human immunodeficiency virus (HIV) status, index tests, reference standard (s) used, number of subjects in each group, and the number of positive and negative test results for each category of subjects.

## Statistical analysis

We computed sensitivity and specificity for either index test from each study and calculated their corresponding 95% confidence intervals (95% CI) using the Clopper-Pearson approach [13]. We used 0.5 as continuity correction for publications reporting zero cell frequencies.

Both Xpert and Ultra assays employ uniform manufacturer-recommended positivity criteria for reporting test results. We therefore used hierarchically structured bivariate random-effects modeling to summarize diagnostic accuracy information from individual studies [14]. As a preliminary analysis, we summarized data separately for studies reporting on diagnostic accuracy of Xpert or Ultra, using either mycobacterial culture or composite criteria as reference standard. We used coupled forest plots and summary receiver operating characteristic (SROC) curves for graphical analysis [15]. This provided us broad indicators for differences in diagnostic performance between Xpert and Ultra from different sets of studies. Since direct comparisons of two index tests conducted within each study are superior to indirect comparisons of the same tests from different studies, we then identified publications reporting on the diagnostic accuracy of both Xpert and Ultra in the same study participants [16]. We anticipated only a small number of such publications and attempted a formal comparison only if

three or more studies provided such paired diagnostic accuracy data [16]. For this, we performed meta-regression through a hierarchical summary receiver operating characteristic (HSROC) model that assessed the influence of type of test (Xpert or Ultra) as a covariate while assuming symmetric SROC curves [17].

We assessed methodological quality of all included studies using the QUADAS-2 (Quality Assessment of Diagnostic Accuracy Studies, version 2) tool [18]. We subjectively assessed heterogeneity from visual examination of the confidence limits of individual studies and the width of prediction regions of SROC plots. We also used Higgins' inconsistency index ($I^2$) as a measure of between-study heterogeneity and considered it high for $I^2$ values >0.75 [19]. Heterogeneity was further explored through a separate subgroup analysis for each test, if ten or more studies were available for the primary analysis. For this, data was stratified based on prespecified covariates that included study design, TB burden in country of study, TPE prevalence among study participants, study sample size, nature of non-tuberculous pleural effusions (whether transudates included or not), and nature of pleural fluid specimens (fresh or cryopreserved; whether centrifuged or not). Countries were categorized as high TB burden, or otherwise, based on World Health Organization guidelines [20]. We used Deek's funnel plot to assess the publication bias. We graded the overall quality of evidence using GRADE guidelines [21].

Statistical significance was assessed at p <0.05. We used the Stata software (Intercooled Edition 12.0, Stata Corp, Texas, USA) for statistical analysis. We also used the MetaDAS macro in SAS environment (SAS University Edition version 9.4, SAS Institute Inc., North Carolina, USA) for meta-regression [22].

## Results

### Study characteristics

We found 1095 citations through a search of electronic databases and located another two from additional sources (Fig 1). In all we assessed 146 full-text publications in detail against our inclusion criteria, and finally included 74 for our analysis [23–96]. Of these, 64 (86.5%) studies evaluated Xpert alone, five (6.8%) evaluated Ultra alone, and five (6.8%) evaluated both tests concurrently (S1 Table of online supplement). Three (4.1%) of these studies were reported in a language other than English [39, 50, 92]. The number of study subjects varied between 6 and 714. There were five (6.8%) studies with a case-control design [23, 31, 35, 43, 92]. In all, 45 (60.8%) studies reported their data from high TB burden countries (S1 Table of online supplement). One (1.4%) study was conducted exclusively in HIV seropositive patients [43], while seven (9.5%) others included a variable number of such subjects [37, 38, 42, 44, 53, 70, 89]. There were no HIV seropositive patients in thirteen (17.6%) publications [30, 47, 63, 73, 74, 76, 78, 79, 83, 84, 88, 92, 96], while the remaining did not provide any information. Ten investigators thawed cryopreserved fluid samples for their tests [23, 31, 35, 42, 60, 66, 73, 78, 80, 84]. Pleural fluid was concentrated by centrifugation in 32 studies prior to Xpert/Ultra assay [23–25, 27, 30, 33–35, 37, 38, 43, 44, 46, 48–51, 53, 58, 59, 65, 72–74, 76, 78, 85, 88, 89, 91, 94, 95]. Most investigators (51, 68.9%) used mycobacterial culture as reference standard for diagnosing TPE, while 33 used a composite reference standard (S1 Table of online supplement). Ten (13.5%) of these studies provided results by both criteria [25, 33, 35, 46, 62, 71, 73, 74, 84, 88]. A variable and wide range of clinical, laboratory and outcome parameters were used in varying combinations to define the composite reference standards. Four studies reported having included transudative pleural effusions in the non-tuberculous group [42, 45, 63, 66].

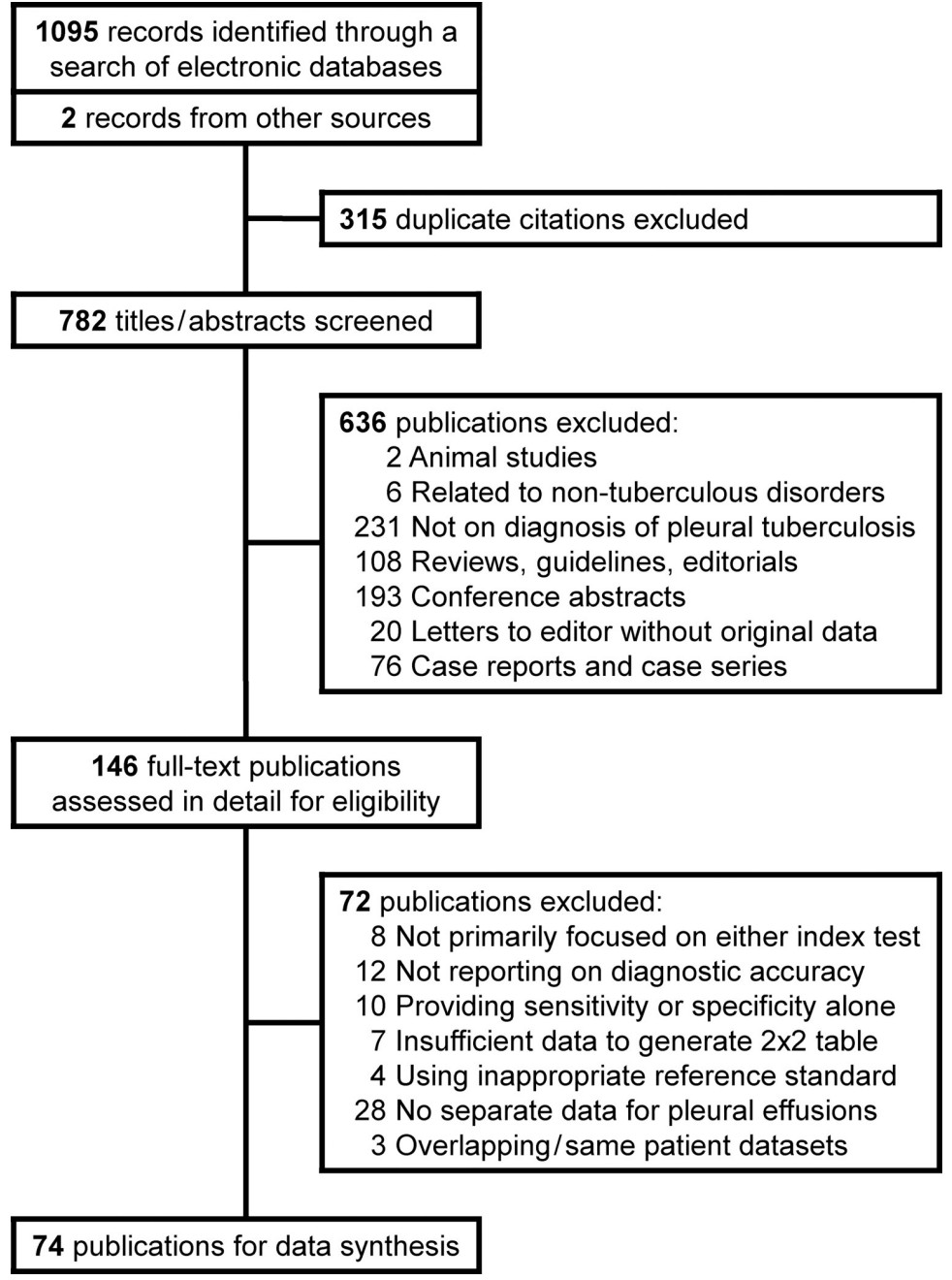

**Fig 1. Study selection process.**

Eleven (14.9%) studies exhibited some risk of bias across one or more QUADAS-2 domains (Fig 2). Thirty (40.5%) studies also showed applicability concerns in one or more QUADAS-2 domains (Fig 2), mostly because the index tests were not conducted strictly as recommended. S2 Table of online supplement summarizes the diagnostic accuracy estimates computed from various studies.

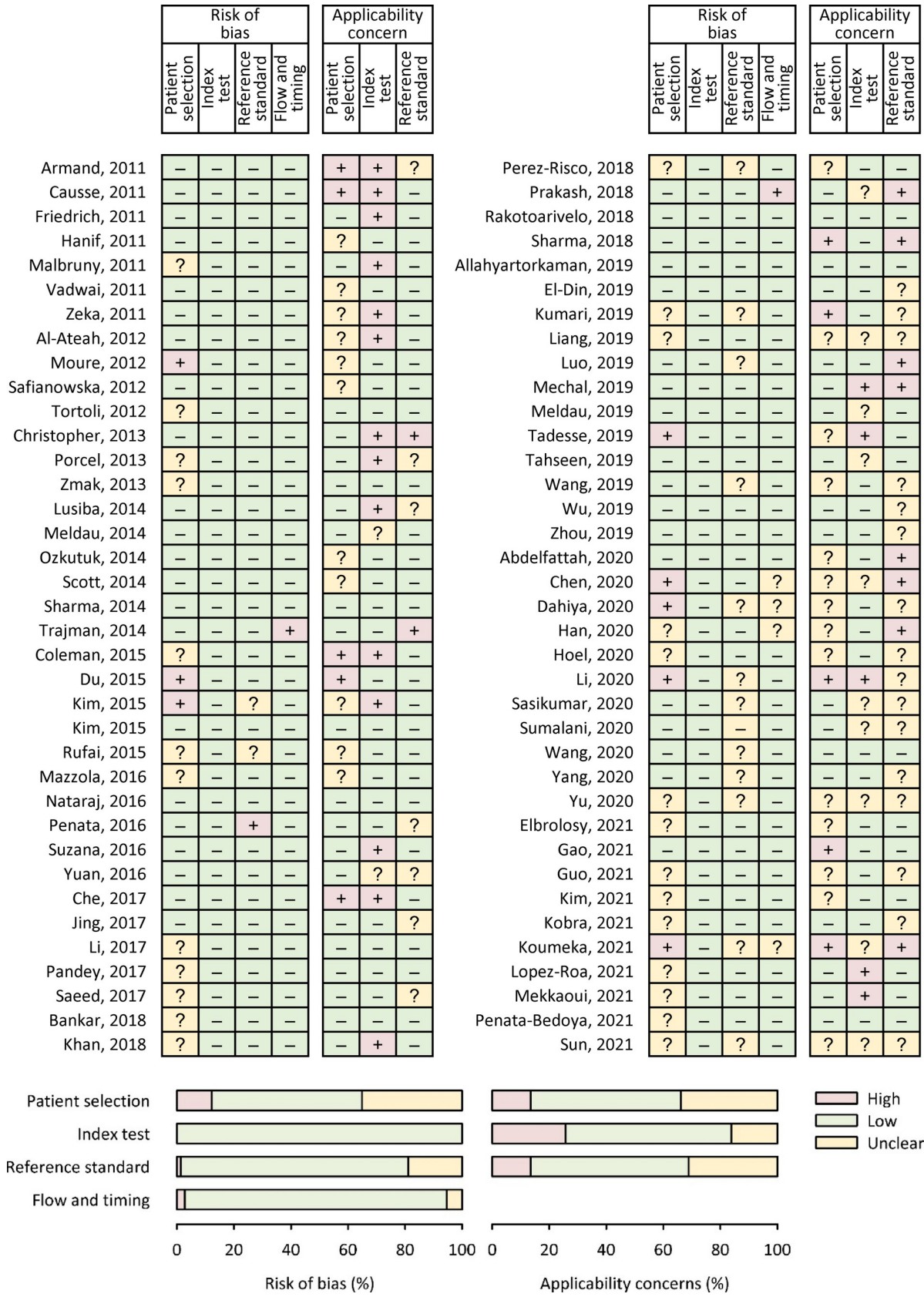

**Fig 2. Risk of bias and applicability concerns summary.**

## Diagnostic accuracy of individual tests

Forty-five studies, with 1203 TPE patients and 5288 patients of other effusions, evaluated Xpert in pleural fluid using mycobacterial culture as reference standard. Xpert sensitivity for TPE diagnosis ranged widely between zero and 1.00 ($I^2$ 82.1%), and specificity between 0.87 and 1.00 ($I^2$ 85.1%) (S1 Fig of online supplement). The summary sensitivity across studies was 0.52 (95% CI 0.43–0.60), and specificity was 0.99 (95% CI 0.97–0.99). The summary positive likelihood ratio (PLR), negative likelihood ratio (NLR) and diagnostic odds ratio (DOR) estimates were 39.10 (95% CI 19.96–76.60), 0.49 (95% CI 0.41–0.59), and 79.98 (95% CI 37.82–169.12) respectively. The SROC curve was placed toward the desirable upper left corner of the plot area, and the 95% prediction region was wide, suggesting between-study heterogeneity (S2 Fig of online supplement). Subgroup analysis did not suggest any obvious influence of the prespecified covariates on heterogeneity, except that case-control studies showed considerable homogeneity in specificity estimates, and use of cryopreserved specimens was associated with lesser diagnostic accuracy but better homogeneity (S3 Table of online supplement). There was no publication bias.

Additionally, nine studies, including 194 TPE patients and 747 patients of other effusions, evaluated Ultra in pleural fluid using mycobacterial culture as reference standard. Sensitivity of Ultra for diagnosis of TPE ranged widely between zero and 1.00 ($I^2$ 80.0%), and specificity between 0.68 and 1.00 ($I^2$ 92.1%) (S3 Fig of online supplement). The summary sensitivity across studies was marginally better than Xpert (0.68, 95% CI 0.55–0.79), and specificity was marginally inferior than Xpert (0.97, 95% CI 0.97–0.99) (Table 1). The summary positive likelihood ratio (PLR), negative likelihood ratio (NLR) and diagnostic odds ratio (DOR) estimates were 27.25 (95% CI 4.56–162.99), 0.33 (95% CI 0.22–0.47), and 83.79 (95% CI 15.53–452.06) respectively. The SROC curve was placed toward the desirable upper left corner of the plot area, and the 95% prediction region was wide, indicating between-study heterogeneity (S2 Fig of online supplement). We did not perform subgroup analysis due to small number of studies. There was no publication bias.

Thirty-five studies, with 2249 TPE patients and 2033 patients of other effusions, assessed Xpert in pleural fluid against a composite reference standard. Xpert sensitivity for detecting TPE ranged widely between zero and 0.71 ($I^2$ 81.5%), and specificity between 0.95 and 1.00 ($I^2$ 37.6%) (S1 Fig of online supplement). The summary sensitivity across studies was 0.21 (95% CI 0.17–0.26), and specificity was 1.00 (95% CI 0.99–1.00). The summary positive likelihood ratio (PLR), negative likelihood ratio (NLR) and diagnostic odds ratio (DOR) estimates were 110.97 (95% CI 25.70–479.06), 0.79 (95% CI 0.74–0.84), and 140.95 (95% CI 32.32–614.74) respectively. The SROC curve was placed close to the left margin of the plot area, and the 95% prediction region was relatively narrow, suggestive of lesser between-study heterogeneity (S2 Fig of online supplement). Subgroup analysis suggested that retrospective studies, studies with less than 100 patients, studies reporting data only from exudative effusions, and studies assaying pleural fluid without centrifugation showed considerable homogeneity in specificity estimates (S3 Table of online supplement). There was no publication bias.

In addition, five studies, with 498 TPE patients and 245 patients of other effusions, assessed Ultra in pleural fluid against a composite reference standard. Sensitivity of Ultra for TPE identification ranged widely between 0.38 and 0.71 ($I^2$ 64.1%), and specificity between 0.90 and 1.00 ($I^2$ 54.8%) (S3 Fig of online supplement). The summary sensitivity across studies was better than Xpert (0.47, 95% CI 0.40–0.55), and specificity was marginally lower than Xpert (0.98, 95% CI 0.95–0.99) (Table 1). The summary positive likelihood ratio (PLR), negative likelihood ratio (NLR) and diagnostic odds ratio (DOR) estimates were 21.88 (95% CI 8.81–54.33), 0.54 (95% CI 0.47–0.62), and 40.68 (95% CI 16.15–102.46) respectively. The SROC curve was

**Table 1. Summary diagnostic accuracy parameters and their comparison.**

| | Xpert MTB/RIF Ultra | Xpert MTB/RIF |
|---|---|---|
| **Independent analysis for each index test** | | |
| 1. Mycobacterial culture as reference standard | | |
| • Number of included studies | 9 | 45 |
| • Summary sensitivity (95% CI) | 0.68 (0.55–0.79) | 0.52 (0.43–0.60) |
| • Summary specificity (95% CI) | 0.97 (0.85–1.00) | 0.99 (0.97–0.99) |
| 2. Composite reference standard | | |
| • Number of included studies | 5 | 35 |
| • Summary sensitivity (95% CI) | 0.47 (0.40–0.55) | 0.21 (0.17–0.26) |
| • Summary specificity (95% CI) | 0.98 (0.95–0.99) | 1.00 (0.99–1.00) |
| **Direct head-to-head comparison of both tests** | | |
| 1. Mycobacterial culture as reference standard | | |
| • Number of included studies | 4 | 4 |
| • Summary sensitivity (95% CI) | 0.78 (0.63–0.87) | 0.42 (0.28–0.59) |
| • Summary specificity (95% CI) | 0.88 (0.56–0.98) | 0.96 (0.82–0.99) |
| • Relative diagnostic odds ratio (95% CI)* | 1.28 (0.65–2.50) | |
| • Relative sensitivity (95% CI)* | 1.83 (1.37–2.46) | |
| • Relative specificity (95% CI)* | 0.91 (0.78–1.06) | |
| 2. Composite reference standard | | |
| • Number of included studies | 5 | 5 |
| • Summary sensitivity (95% CI) | 0.47 (0.40–0.55) | 0.23 (0.18–0.29) |
| • Summary specificity (95% CI) | 0.98 (0.95–0.99) | 0.99 (0.96–1.00) |
| • Relative diagnostic odds ratio (95% CI)* | 1.80 (0.41–7.84) | |
| • Relative sensitivity (95% CI)* | 2.07 (1.70–2.51) | |
| • Relative specificity (95% CI)* | 0.99 (0.97–1.02) | |

* Xpert MTB/RIF Ultra in comparison to Xpert MTB/RIF

CI confidence interval

placed close to the left margin of the plot area, and the 95% prediction region was relatively narrow, suggestive of moderate between-study heterogeneity (S2 Fig of online supplement). Subgroup analysis was not performed due to small number of studies. There was no publication bias.

## Comparative diagnostic accuracy of both tests

Only five studies, all from high TB burden countries, evaluated diagnostic accuracy of both Xpert and Ultra in pleural fluid in the same study population [70, 73, 74, 84, 88]. None had a case-control design. The number of study subjects ranged from 61 to 292. Four of these publications from China provided information for both mycobacterial culture and composite criteria as reference standards [73, 74, 84, 88], and one from South Africa used only composite reference standard [70]. One study used previously archived pleural fluid samples from a biobank [73]. None of the Chinese studies had any HIV seropositive patient, but the South African study reported 14.2% HIV seropositivity rate [70]. No study reported evaluation of any transudative pleural effusion. There was no apparent risk of bias in any study, but the risk of bias in the reference standard domain was not clear for two studies [73, 84].

Four studies, with 155 TPE patients and 458 patients of other effusions, evaluated both Xpert and Ultra in pleural fluid using mycobacterial culture as reference standard [73, 74, 84,

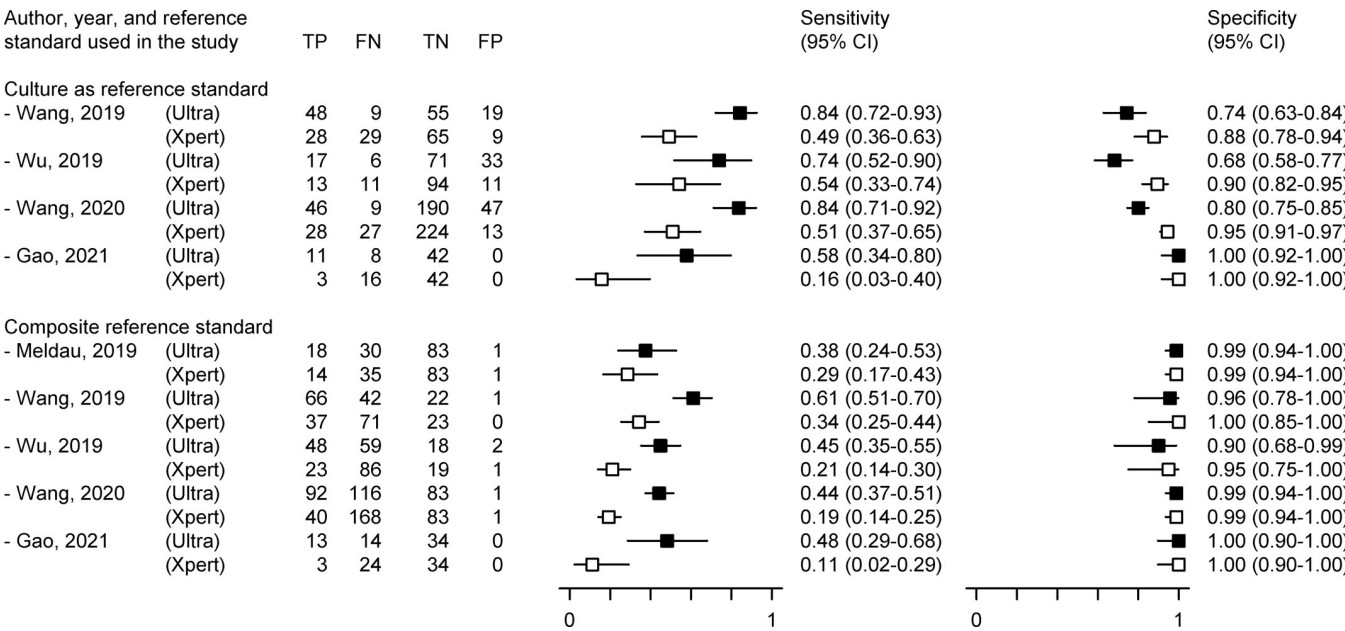

**Fig 3. Coupled forest plot from studies on diagnostic accuracy of pleural fluid Xpert MTB/RIF and Xpert MTB/RIF Ultra in the same patient population.** Individual sensitivity and specificity estimates for diagnosing tuberculous pleural effusion are derived from data on true positives (TP), false negatives (FN), true negatives (TN), and false positives (FP), and are represented by solid and hollow squares for Xpert MTB/RIF Ultra and Xpert MTB/RIF respectively. Horizontal lines depict 95% confidence interval.

88]. All studies showed a higher sensitivity, and lower or equal specificity, for Ultra (Fig 3). On meta-regression, when compared to Xpert, testing with Ultra resulted in higher summary sensitivity (0.78, 95% CI 0.63–0.87 vs. 0.42, 95% CI 0.28–0.59) but lower summary specificity (0.88, 95% CI 0.56–0.98 vs. 0.96, 95% CI 0.82–0.99). The corresponding SROC plots for the two assays did not overlap, and the curve for Ultra was located more towards the upper left corner of SROC space (Fig 4), implying that Ultra was consistently better than Xpert in diagnosing TPE across the whole range of data from the studies analyzed. However, the 95% confidence and prediction ellipses around both the summary estimates were wide and overlapping (Fig 4), implying significant heterogeneity. The relative diagnostic odds ratio (RDOR, a summary measure of relative accuracy) for Ultra was 1.28 (95% CI 0.65–2.50), suggestive of no significant difference in summary diagnostic accuracy between the two tests. However, Ultra showed significantly better sensitivity (relative sensitivity 1.83, 95% CI 1.37–2.46), but a similar specificity (Table 1).

Five studies, with 501 TPE patients and 245 patients of other effusions, evaluated both Xpert and Ultra in pleural fluid using a composite reference standard [70, 73, 74, 84, 88]. All studies showed a higher sensitivity, and lower or equal specificity, for Ultra (Fig 3). On meta-regression, when compared to Xpert, testing with Ultra resulted in higher summary sensitivity (0.47, 95% CI 0.40–0.55 vs. 0.23, 95% CI 0.18–0.29) but lower summary specificity (0.98, 95% CI 0.95–0.99 vs. 0.99, 95% CI 0.96–1.00). The corresponding SROC plots for the two assays were positioned close to each other but did not overlap, and the curve for Ultra was located more towards the upper left corner of SROC space (Fig 4), implying that Ultra was marginally better than Xpert in diagnosing TPE across the whole range of data from the studies analyzed. However, the 95% confidence and prediction ellipses around both the summary estimates were medium-sized and overlapping (Fig 4), implying moderate heterogeneity. The RDOR for Ultra was 1.80 (95% CI 0.41–7.84), suggestive of no significant difference in summary

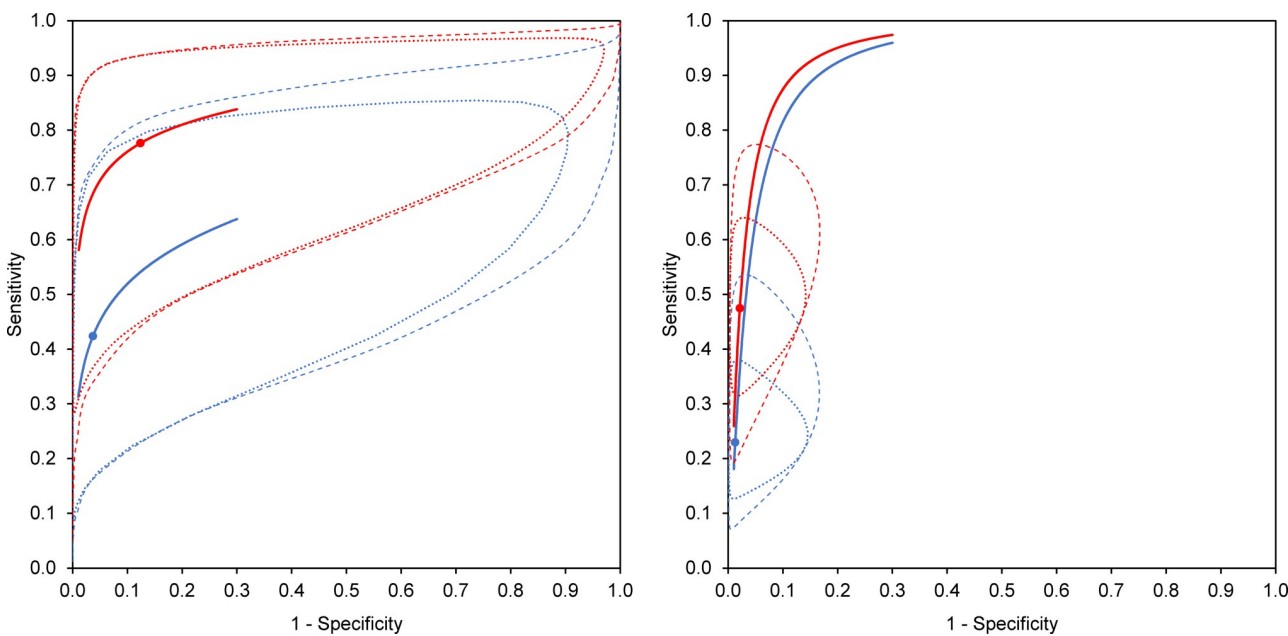

**Fig 4. Comparison of summary points and hierarchical summary receiver operating characteristic plots for studies evaluating both pleural fluid Xpert MTB/RIF (blue) and Xpert MTB/RIF Ultra (red), using mycobacterial culture (left panel) and composite criteria (right panel) as reference standard for diagnosing tuberculous pleural effusion.** Summary diagnostic accuracy points are depicted by solid circles. The dotted ellipses characterize the 95% confidence region around these summary estimates, while the dashed ellipses represent the 95% prediction region (area within which one is 95% certain the results of a new study will lie).

diagnostic accuracy between the two tests. However, Ultra showed significantly better sensitivity (relative sensitivity 2.07, 95% CI 1.70–2.51), but a similar specificity (Table 1).

## Grading of evidence

Based on the summary diagnostic accuracy estimates derived from comparative studies, we projected the relative yield of the two index tests at low (5%), and high (50%) pre-test probability of TPE (Table 2). When using mycobacterial culture as reference standard in a low prevalence setting, the extra TPE patients identified through Ultra were overshadowed by a far greater number of false positive test results. Such disagreement was, however, not noted in a high TPE prevalence setting, or with comparisons using a composite reference standard (Table 2). This discrepancy was considered to suggest imprecision in relative specificity estimates among studies using mycobacterial culture as the reference standard. In view of this, and the wide confidence intervals for true negative and false positive estimates, we downgraded the level of certainty of evidence to 'moderate' for specificity comparisons using culture as reference standard. Other comparisons were considered to provide high certainty of evidence (Table 2).

## Discussion

We reviewed 74 publications reporting on the diagnostic accuracy of pleural fluid Xpert or Ultra in TPE. In independent analyses, both tests showed low-to-moderate summary sensitivity and high summary specificity. Ultra had higher summary sensitivity than Xpert, both when mycobacterial culture (0.68 from nine studies vs. 0.52 from 45 studies) and composite criteria (0.47 from five studies vs. 0.21 from 35 studies) were used as the reference standard. Summary specificity was marginally lower for Ultra. On direct comparative analysis through HSROC

**Table 2. Summary of findings from studies comparing both pleural fluid Xpert and Ultra assays for diagnosing tuberculous pleural effusion in the same patient population.**

| Test result | Number of subjects (number of studies) | Number of results per 1000 patients tested (95% confidence interval) | | | | Risk of bias Inconsistency Indirectness Publication bias | Imprecision | Certainty of the evidence |
|---|---|---|---|---|---|---|---|---|
| | | 5% prevalence of tuberculosis | | 50% prevalence of tuberculosis | | | | |
| | | Ultra | Xpert | Ultra | Xpert | | | |
| Mycobacterial culture as reference standard | | | | | | | | |
| True positives | 155 (4) | 39 (32 to 44) | 21 (14 to 29) | 388 (317 to 437) | 212 (138 to 294) | Not serious | Not serious | HIGH |
| | | 18 more with Ultra | | 176 more with Ultra | | | | |
| False negatives | | 11 (6 to 18) | 29 (21 to 36) | 112 (63 to 183) | 288 (206 to 362) | | | |
| | | 18 fewer with Ultra | | 176 fewer with Ultra | | | | |
| True negatives | 458 (4) | 833 (529 to 927) | 915 (778 to 944) | 438 (278 to 488) | 482 (409 to 497) | Not serious | Serious [a] | MODERATE |
| | | 82 fewer with Ultra | | 44 fewer with Ultra | | | | |
| False positives | | 117 (23 to 421) | 35 (6 to 172) | 62 (12 to 222) | 18 (3 to 91) | | | |
| | | 82 more with Ultra | | 44 more with Ultra | | | | |
| Composite reference standard | | | | | | | | |
| True positives | 501 (5) | 24 (20 to 28) | 11 (9 to 15) | 237 (200 to 275) | 115 (89 to 146) | Not serious | Not serious | HIGH |
| | | 13 more with Ultra | | 122 more with Ultra | | | | |
| False negatives | | 26 (22 to 30) | 39 (35 to 41) | 263 (225 to 300) | 385 (354 to 411) | | | |
| | | 13 fewer with Ultra | | 122 fewer with Ultra | | | | |
| True negatives | 245 (5) | 930 (902 to 942) | 938 (913 to 946) | 489 (475 to 496) | 494 (480 to 498) | Not serious | Not serious | HIGH |
| | | 8 fewer with Ultra | | 5 fewer with Ultra | | | | |
| False positives | | 20 (8 to 48) | 12 (4 to 37) | 11 (4 to 25) | 6 (2 to 20) | | | |
| | | 8 more with Ultra | | 5 more with Ultra | | | | |

[a] Wide confidence limits for estimates, and a disproportionally large increase in number of false positives, more so in a low tuberculosis prevalence setting

meta-regression from studies with paired datasets, Ultra had a RDOR of 1.28 and 1.80 respectively when compared to Xpert, using culture (four studies) and composite criteria (five studies) as reference standard. Our results suggest Ultra to be the better diagnostic investigation for TPE.

The summary diagnostic accuracy estimates computed by us, individually for both pleural fluid Xpert and Ultra, are largely similar to those reported by recent meta-analyses [5, 9]. A direct comparative analysis of studies reporting paired diagnostic accuracy data is preferred to deriving indirect inferences from different meta-analyses on individual tests, as the former removes confounding due to differences in study methodology and patient characteristics [97]. A recent Cochrane review did not perform a direct comparative analysis due to paucity of studies providing concurrent information on both pleural fluid Xpert and Ultra for the same patients [5]. Another review identified four studies providing paired data on pleural fluid Xpert and Ultra, but reported only the individual summary diagnostic accuracy estimates separately for each test without specifying the reference standard [98].

What are the clinical implications of our study? The positioning of HSROC plots, as well as the numerical information for summary estimates from studies providing paired data, suggests

pleural fluid Ultra to be a better diagnostic marker for TPE than pleural fluid Xpert. This information is likely to influence current algorithms for evaluating patients with pleural effusion in whom TB is considered as one of the possible etiologies, especially once the Ultra kits become more widely available. Our estimates suggest that using Ultra might paradoxically increase false positive rates in low TB prevalence settings if mycobacterial culture is considered as the reference standard. This is not the case if composite criteria are employed as the reference standard. Notably, all studies included for our comparative meta-analyses were conducted in high TB burden countries. Neither mycobacterial culture nor composite criteria can be considered an ideal reference standard. Since culture requires a much higher viable mycobacterial load than nucleic acid amplification assays, it may be possible that some of the extra cases identified by Ultra (and categorized as false positives) actually represent those patients whose diagnosis was missed by the definitive reference standard. The lower limit of detecting mycobacterial genetic material in pleural fluid is further approximately ten-fold lower for Ultra as compared to Xpert [70]. This might be advantageous for diagnosing TPE, a paucibacillary condition. On the other hand, using composite criteria lowers the precision in picking up true TPE, and the problem is further compounded by the fact that different investigators used variable composite criteria to define TPE without providing additional information on treatment outcomes stratified by culture or Xpert/Ultra results or by pleural fluid characteristics. From a purely medical perspective, physicians tend to consider several clinical and laboratory parameters while assigning a presumptive diagnosis of TPE. Moreover, culture reporting takes time, and results are often not available while deciding on initiation of anti-tubercular treatment.

The main strengths of our analysis are a larger sample size of paired data on the two index tests, and the use of hierarchical models for formal test comparison, allowing us to generate robust comparative diagnostic accuracy estimates. Our evaluation also has few limitations. The studies reviewed herein showed substantial heterogeneity. Only a few studies enrolled patients with exudative pleural effusions only. As TPE is not a diagnostic consideration in transudative effusions, several studies may have reported a spuriously higher specificity. We summarized and compared the diagnostic accuracy of Xpert and Ultra as isolated investigations, but cannot judge if their concurrent use with results of other diagnostic tests can further expand their role in routine clinical decision-making. Nearly all studies describing role of both Ultra and Xpert on the same patient dataset were performed in a single country, precluding the generalizability of our findings to other locations.

## Conclusion

In summary, the results from our meta-analysis suggest that pleural fluid Ultra assay provides superior diagnostic accuracy over Xpert assay for diagnosing TPE, mainly because of its higher sensitivity. We propose that pleural fluid Ultra should be used as a primary diagnostic biomarker while evaluating patients with suspected TPE, especially in high TB prevalence settings. More information, especially on Ultra's positioning in any diagnostic algorithm evaluating pleural effusions and it utility when combined with other clinical and laboratory data, is needed to fully characterize the added advantage of Ultra in different countries and settings.

## Supporting information

**S1 Checklist. PRISMA-DTA checklist.**
(PDF)

**S1 Table. Characteristics of studies included in data synthesis.**
(PDF)

**S2 Table. Diagnostic accuracy estimates from included studies.**
(PDF)

**S3 Table. Evaluation of factors affecting individual summary diagnostic accuracy estimates from studies on pleural fluid Xpert MTB/RIF assay.**
(PDF)

**S1 Fig. Forest plots of studies evaluating sensitivity and specificity of pleural fluid Xpert MTB/RIF assay in diagnosing tuberculous pleural effusion.** Solid squares indicate individual study estimates, and horizontal lines represent corresponding 95% confidence limits.
(PDF)

**S2 Fig. Summary receiver operating characteristic (SROC) curves from bivariate models summarizing diagnostic performance of pleural fluid Xpert MTB/RIF using culture (top left) or composite criteria (top right) as reference standard, and pleural fluid Xpert MTB/ RIF Ultra using culture (bottom left) or composite criteria (bottom right) as reference standard.** Each individual study on diagnosis of tuberculous pleural effusion is represented by an open circle, whose size is proportional to the inverse standard error of sensitivity and specificity. The square represents the summary estimate of test accuracy, with the surrounding dashed zone outline denoting the 95% confidence region around this estimate. The outer dotted zone represents the 95% prediction region (area within which one is 95% certain the results of a new study will lie).
(PDF)

**S3 Fig. Forest plots of studies evaluating sensitivity and specificity of pleural fluid Xpert MTB/RIF Ultra in diagnosing tuberculous pleural effusion.** Solid squares indicate individual study estimates, and horizontal lines represent corresponding 95% confidence limits.
(PDF)

## Author Contributions

**Conceptualization:** Ashutosh Nath Aggarwal.

**Data curation:** Ashutosh Nath Aggarwal, Ritesh Agarwal.

**Formal analysis:** Ashutosh Nath Aggarwal, Ritesh Agarwal, Sahajal Dhooria, Kuruswamy Thurai Prasad, Inderpaul Singh Sehgal, Valliappan Muthu.

**Methodology:** Ashutosh Nath Aggarwal, Ritesh Agarwal, Sahajal Dhooria, Kuruswamy Thurai Prasad, Inderpaul Singh Sehgal, Valliappan Muthu.

**Project administration:** Ashutosh Nath Aggarwal.

**Supervision:** Ashutosh Nath Aggarwal, Ritesh Agarwal.

**Validation:** Ashutosh Nath Aggarwal, Ritesh Agarwal, Sahajal Dhooria, Kuruswamy Thurai Prasad, Inderpaul Singh Sehgal, Valliappan Muthu.

**Visualization:** Ashutosh Nath Aggarwal, Ritesh Agarwal.

**Writing – original draft:** Ashutosh Nath Aggarwal, Ritesh Agarwal, Sahajal Dhooria, Kuruswamy Thurai Prasad, Inderpaul Singh Sehgal, Valliappan Muthu.

**Writing – review & editing:** Ashutosh Nath Aggarwal, Ritesh Agarwal, Sahajal Dhooria, Kuruswamy Thurai Prasad, Inderpaul Singh Sehgal, Valliappan Muthu.

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
