## [Decision Letter · Decision Letter 0]

20 Apr 2022

PONE-D-22-01521Xpert MTB/RIF Ultra versus Xpert MTB/RIF for diagnosis of tuberculous pleural effusion: a systematic review and comparative meta-analysisPLOS ONE

Dear Dr. Aggarwal,

Thank you for submitting your manuscript to PLOS ONE. After careful consideration, we feel that it has merit but does not fully meet PLOS ONE’s publication criteria as it currently stands. Therefore, we invite you to submit a revised version of the manuscript that addresses the points raised during the review process.

We look forward to receiving your revised manuscript.

Kind regards,

Juraj Ivanyi

Academic Editor

PLOS ONE

Journal Requirements:

2. Please amend your list of authors on the manuscript to ensure that each author is linked to an affiliation. Authors’ affiliations should reflect the institution where the work was done (if authors moved subsequently, you can also list the new affiliation stating “current affiliation:….” as necessary).

Reviewers' comments:

Reviewer's Responses to Questions

**Comments to the Author**

1. Is the manuscript technically sound, and do the data support the conclusions?

Reviewer #1: Yes

Reviewer #2: Yes

2. Has the statistical analysis been performed appropriately and rigorously? 

Reviewer #1: Yes

Reviewer #2: Yes

3. Have the authors made all data underlying the findings in their manuscript fully available?

Reviewer #1: Yes

Reviewer #2: Yes

4. Is the manuscript presented in an intelligible fashion and written in standard English?

Reviewer #1: Yes

Reviewer #2: Yes

5. Review Comments to the Author

Reviewer #1: This meta-analysis updates previous meta-analyses to help decide the relative benefits of Xpert MTB/RIF and Xpert Ultra in making a correct diagnosis of a tuberculous pleural effusion (TPE).

Whilst culture of Mycobacterium tuberculosis (Mtb) is undoubtedly the gold standard, as the authors note, culture of Mtb is rare in TPE as the bacterial load is low and the fluid accumulates as a result of the delayed hypersensitivity reaction to mycobacterial antigens. In culture-negative TPE, the probability of a correct diagnosis varies significantly from histopathological confirmation, through lymphocyte-predominant effusions with a high protein and low glucose to empirical treatment in a setting where tuberculosis is common. A composite criterion of TPE should therefore account for the numbers in each of the three above groups. From the methods section, it would seem that the subgroup analysis accounted for variables other than these clinical groups of different pre-test probabilities of TPE (lines 190-3).

Alternatively, the authors could state more clearly that, in those with a culture-negative TPE, the positive PCR tests correlated better with a good TB treatment outcome than would be expected by chance. Papers reporting such data would require absence of a blood neutrophilia and follow-up for 1-5 years after treatment completion (untreated TPE results in post-primary TB usually within 5 years).

The abstract should report the I-squared value, indicating that most of the differences were due to heterogeneity in the studies.

The introduction should note the prevalence of IS1081 in mycobacterial species other then Mtb sensu stricto. This will provide a background for discussion of the lower specificity of Xpert Ultra (M bovis, BCG, M smegmatis etc., and also related to the restriction enzyme used).

The methods should indicate the nature of the composite score (see above comments0. Were different criteria weighted or given equal scores? Line 364 implies that they had data related to the different composite scores.

The conclusion should include the “more information” required to better evaluate the Ultra test (also noted above in the 2nd and 3rd paragraphs).

The supplemental figure 1 (risk of bias…) is of such significance that it should be included in the main text.

Typographical errors

Line 116: Health not health

Reviewer #2: In this manuscript, the authors aim to compare the accuracy of two tests used to diagnose TPE based on biomolecular techniques. The paper is well written and structured. It fits the PRISMA checklist for meta-analysis and systematic reviews.

The results obtained are summarized in table 2 (even if in the text, this table is called table 3).

The ULTRA test shows higher specificity and almost the same sensibility compared to TXPERT. But in a low incidence of tuberculosis cases, it looks not accurate due to the high ratio of false positives if the microbiological parameter for validation is used. In the discussion the authors provide an interesting analysis of the obtained results.

In my opinion, the paper can be published almost as it is. I have only two questions for the authors.

1) In lines 321 and 326, about the summary of findings, the text refers to table three, but these data are showed in table two; can the authors check?

2) In figure 1, the workflow of analysis display that the authors used 74 publications for data synthesis, but in lines 229 and 253, the authors state the use of 45 publication with Mtb culture as the reference standard and 35 publications with composite reference standard respectively. The same numbers, 45 and 35, are also reported in supplemental table three. Can the authors better specify if they used 74 or 80 publications for the analysis?

6. PLOS authors have the option to publish the peer review history of their article (what does this mean?). If published, this will include your full peer review and any attached files.

Reviewer #1: No

Reviewer #2: No

---

## [Author Response · Author response to Decision Letter 0]

25 Apr 2022

Responses to comments from Reviewer #1: 

Comment:

Alternatively, the authors could state more clearly that, in those with a culture-negative TPE, the positive PCR tests correlated better with a good TB treatment outcome than would be expected by chance. Papers reporting such data would require absence of a blood neutrophilia and follow-up for 1-5 years after treatment completion (untreated TPE results in post-primary TB usually within 5 years).

Response:

We had focussed sorely on diagnsotic accuracy information and not on treatment oucomes. No study provided detailed information on treatment oucomes stratified by culture or PCR results, or by pleural fluid characterestics. This has been added as a limitation in Discussion (para 3, lines 365-366).

Comment:

The abstract should report the I-squared value, indicating that most of the differences were due to heterogeneity in the studies.

Response:

We have provided the I-squared values in abstract (lines 76-81).

Comment:

The introduction should note the prevalence of IS1081 in mycobacterial species other then Mtb sensu stricto. This will provide a background for discussion of the lower specificity of Xpert Ultra (M bovis, BCG, M smegmatis etc., and also related to the restriction enzyme used).

Response:

Both IS6110 and IS1081 sequences are present in in M. tuberculosis complex, but not in other mycobacteria. This has now been mentioned in Introduction (para 2, lines 114-115).

Comment:

The methods should indicate the nature of the composite score (see above comments). Were different criteria weighted or given equal scores? Line 364 implies that they had data related to the different composite scores.

Response:

A variable and wide range of clinical, laboratory and outcome parameters were used in varying combinations to define the composite reference standards, and it was not possible to characterize or summarize this information further. We highlighted this in Discussion (para 3, lines 365-366)

Comment:

The conclusion should include the “more information” required to better evaluate the Ultra test (also noted above in the 2nd and 3rd paragraphs).

Response:

As suggested, we have added text to the last line in Conclusion (lines 386-387).

Comment:

The supplemental figure 1 (risk of bias…) is of such significance that it should be included in the main text.

Response:

We have now included the figure on risk of bias and applicability concerns summary as Fig 2 in the main manuscript, and renumbered the subsequent figures in main manuscript and online supplement accordingly.

Comment:

Typographical errors. Line 116: Health not health

Response:

We apologise for this inadvertant error, and the same has now been corrected (line 118).

Responses to comments from Reviewer #2: 

Comment:

In lines 321 and 326, about the summary of findings, the text refers to table three, but these data are showed in table two; can the authors check?

Response:

We apologise for this typographical error related to citation for Table 2, and the same has now been corrected in Results (last para, lines 319, 322, 327).

Comment:

In figure 1, the workflow of analysis display that the authors used 74 publications for data synthesis, but in lines 229 and 253, the authors state the use of 45 publication with Mtb culture as the reference standard and 35 publications with composite reference standard respectively. The same numbers, 45 and 35, are also reported in supplemental table three. Can the authors better specify if they used 74 or 80 publications for the analysis?

Response:

There were a few studies that provided data for both compositie and culture-based reference standards, and hence the overlap in numbers. This has already been clarified while reporting the study characterestics (Results, para 1, lines 218-220).

---

## [Editor Report · Decision Letter 1]

1 May 2022

Xpert MTB/RIF Ultra versus Xpert MTB/RIF for diagnosis of tuberculous pleural effusion: a systematic review and comparative meta-analysis

PONE-D-22-01521R1

Dear Dr. Aggarwal,

We’re pleased to inform you that your manuscript has been judged scientifically suitable for publication and will be formally accepted for publication once it meets all outstanding technical requirements.

Kind regards,

Juraj Ivanyi

Academic Editor

PLOS ONE
---

## [Editor Report · Acceptance letter]

1 Jul 2022

PONE-D-22-01521R1 

Xpert MTB/RIF Ultra versus Xpert MTB/RIF for diagnosis of tuberculous pleural effusion: a systematic review and comparative meta-analysis 

Dear Dr. Aggarwal:

I'm pleased to inform you that your manuscript has been deemed suitable for publication in PLOS ONE. Congratulations! Your manuscript is now with our production department. 

Kind regards, 

on behalf of

Dr. Juraj Ivanyi 

Academic Editor

PLOS ONE